# Single-Walled Carbon Nanotubes (SWCNTs) as Solid-Contact in All-Solid-State Perchlorate ISEs: Applications to Fireworks and Propellants Analysis

**DOI:** 10.3390/s19122697

**Published:** 2019-06-14

**Authors:** Saad S. M. Hassan, Ahmed Galal Eldin, Abd El-Galil E. Amr, Mohamed A. Al-Omar, Ayman H. Kamel

**Affiliations:** 1Chemistry Department, Faculty of Science, Ain Shams University, Abbasia, Cairo 11566, Egypt; ahmeddna2006@yahoo.com; 2Pharmaceutical Chemistry Department, Drug Exploration & Development Chair (DEDC), College of Pharmacy, King Saud University, Riyadh 11451, Saudi Arabia; malomar1@ksu.edu.sa; 3Applied Organic Chemistry Department, National Research Centre, Dokki, Giza 12622, Egypt

**Keywords:** perchlorate, solid-contact ISEs, SWCNTs, potentiometric sensors, indium-porphyrin, fireworks and propellants

## Abstract

Herein, we present reliable, robust, stable, and cost-effective solid-contact ion-selective electrodes (ISEs) for perchlorate determination. Single-walled carbon nanotubes (SWCNTs) were used as solid-contact material and indium (III) 5, 10, 15, 20-(tetraphenyl) porphyrin chloride (In^III^-porph) as an ion carrier. The sensor exhibited an improved sensitivity towards ClO_4_^−^ ions with anionic slope of −56.0 ± 1.1 (R^2^ = 0.9998) mV/decade over a linear range 1.07 × 10^−6^ – 1.0 × 10^−2^ M and detection limit of 1.8 × 10^−7^ M. The short-term potential stability and the double-layer capacitance were measured by chronopotentiometric and electrochemical impedance spectroscopy (EIS) measurements, respectively. The sensor is used for ClO_4_^−^ determination in fireworks and propellant powders. The results fairly agree with data obtained by ion chromatography.

## 1. Introduction

Perchlorate ions can be found due to either natural processes or as a result of human activities. These ions are characterized by their high solubility, high salvation capacity, and high reduction potential in water. These properties make perchlorate ions both chemically stable and risky towards human health [1]. Exposure to perchlorate can affect the thyroid gland function. It interferes with the uptake of iodide and the production of thyroid hormone. Standards were set by official agencies such as medical, clinical, or environmental laboratories in order to face these health threats [2]. Perchlorate salts were integrated in industry as rocket solid propellants and military explosives. In addition, they have been used as initiators, detonators, and blasting agents. Many aerospace programs in addition to more than 40 different weapon systems are based on perchlorate. Salts of perchlorate are used in the manufacturing of fireworks, flares, and coin-cell batteries. Moreover, they can be used as an automobile airbag initiator, in pyrotechnic devices, finishing leather, and in electronic tubes [3].

Various analytical techniques have been used for perchlorate determination [4,5,6,7,8,9,10,11,12,13,14]. Among of these methods are titrimetry [6], gravimetry [7], dye extraction spectrophotometry [8,9], atomic absorption spectrometry (AAS) [10,11], ion chromatography (IC) [12,13], and mass spectrometry based on electrospray ionization [14]. The main disadvantages of these analytical methods are the poor sensitivity and low selectivity [6,7], the high cost instrumentation [12,13,14], and the extensive sample pretreatment [8,9].

Potential-based sensors or the so-called “ion selective electrodes” (ISEs) have been extensively introduced in different analytical applications, such as in clinical analysis [15,16,17,18,19], environmental monitoring [20,21,22,23,24,25], pharmaceutical analysis [26,27,28,29,30,31], and quality control criteria [32]. This class of analytical devices is characterized by their low cost, high reliability and validity, and ease of operation.

Solid-contact ion-selective electrodes (SC-ISEs) as a different generation of ISEs are characterized by their suitable storage and servicing, ease of miniaturization, and high solidity [33]. The presence of the “blocked” interface between the electronic conductor and ion-selective membrane (ISM) is removed by the insertion of solid-contact materials, such as carbon nano-structures, conducting polymers or nano-noble metals. Signal noises and potential drifts, which can restrict the applications of ISEs are now removed [34]. In the literature, many ISEs have been reported for perchlorate assessment. Most of these electrodes are based on the use of perchlorate/metal chelates ion-association complexes [35,36,37,38,39], quaternary ammonium ions with long chains [40,41,42], and organic dyes [43,44,45]. These electrodes have poor sensitivity towards trace levels of ClO_4_^−^ in presence of other many common anions, such as hydroxide, nitrate, thiocyanate, and iodide. Other reported perchlorate ISEs based on either neutral or charged carriers showed improved selectivity and sensitivity [45,46,47,48,49,50,51,52]. Other ISEs based on surfactant-modified zeolite Y (SMZ) nano-clusters have also been reported for perchlorate determination [53,54]. However, the development of robust and reliable ClO_4_^−^ ISEs with good selectivity and high sensitivity is still a needed request for dealing with samples of small volumes.

In this study, we present a new robust, reliable, sensitive, and cost-effective solid-contact ISE for fast perchlorate determination. The sensor is based on indium-porphyrin ionophore in the sensing membrane and single-walled carbon nanotubes (SWCNTs) as solid-contact material. The structure of SWCNTs contributes to a high double layer capacitance because of their large specific surface area. It also reveals good electric conductivity in addition to its high hydrophobicity. The proposed sensor is used for the assay of ClO_4_^−^ in fireworks and propellant samples.

## 2. Materials and Methods

### 2.1. Reagents

The ionophore indium (III) 5, 10, 15, 20-(tetraphenyl) porphyrin chloride (In^III^-porph) was purchased from PorphyChem SAS (Dijon, France). Tetradodecylammonium tetrakis (4-chlorophenyl) borate (ETH 500), high molecular weight poly (vinyl chloride) (PVC), 2-nitrophenyl octyl ether (o-NPOE), tridodecylmethylammonium chloride (TDMAC), and tetrahydrofuran (THF) were purchased from Fluka AG (Buchs, Switzerland). Single-walled carbon nanotubes (SWCNTs) were purchased from XFnano Materials Tech Co., Ltd. (Nanjing, China).

Aqueous solutions of the reagents and test solutions were prepared with de-ionized bi-distilled water. A stock solution of 0.1 M ClO_4_^−^ was prepared by dissolving in NaClO_4_ and then diluted to working standard solutions with de-ionized bi-distilled water prior to measurements.

### 2.2. Apparatus

“All potentiometric measurements were carried out at 20–21 °C using an Orion-SA 720 pH/meter (MA, USA) in the galvanic cell: Ag/AgCl/(3 M KCl)/0.1 M LiOAc/sample solution//ISE membrane/SWCNTs/glassy carbon electrode (GCE). Selectivity coefficients for the proposed sensor towards ClO_4_^−^ over different common anions were evaluated and calculated by the modified separate solution method (MSSM) [55]. The modified Debye–Hückel equation was employed for calculation of all activity coefficients of the tested ions [56]”.

“Ion chromatography measurements of perchlorate samples were conducted for comparison using a Thermo IC-1100 system equipped with GP50 gradient pump and ED40 electrochemical conductivity cell detector. A Dionex Ion Pac AS-16 separation column (2 × 250 mm^2^), AS16 guard column (2 × 50 mm^2^), 5 × 10^−2^ M NaOH eluent, 0.5 mL flow rate, and 500 μL perchlorate injection volumes were used”.

“Chronopotentiometry and electrochemical impedance spectroscopy (EIS) measurements were carried out using an Autolab Model 2000 potentiostat/galvanostat (Metrohom Instruments, Herisau, Switzerland). A three-electrode configuration cell containing silver/silver chloride (3 M KCl) reference electrode and an auxiliary electrode made from platinum wire was employed. The impedance spectra were measured and recorded at open-circuit potential in 0.01 M NaClO_4_ solution with excitation amplitude of 10 mV and a frequency range of 100 kHz–0.1 Hz”.

### 2.3. Preparation Procedure of SC-ISEs

“The ion-sensing membrane (ISM) is prepared as mentioned previously [57], by dissolving 360 mg of the membrane components in 2.5 mL of THF: (In^III^-porph) (1 wt %), TDMAC (1 wt %), o-NPOE (49 wt %) and PVC (49.0 wt %). Using sonication, degassing for the membrane cocktail is done for 10 min. The solid-contact ISEs were fabricated as follows: (1) Glassy carbon electrode (GCE) was firstly polished with 0.3 µm γ-Al_2_O_3_ slurries, rinsed with water, sonicated for 10 min in ethanol and then, dried with ethanol. The resulting GCE was placed into a piece of matched PVC tubing at its distal end. (2) Mixture of 20 mg of ETH 500 and 2 mg of SWCNTs were spread onto the electrode surface, heated by an infrared lamp for 10 s till complete melting of the ETH 500. The mixture is then left to cool forming a uniform composite layer that is strongly adhered to the surface of GCE. (3) One-hundred microliters of the membrane cocktail was drop-cast onto the transducer layer and left to dry for 2 h. The GC/ClO_4_^−^-ISEs were prepared by the previously mentioned steps without using SWCNTs. The ClO_4_^−^-ISEs were firstly conditioned in 10^−3^M ClO_4_^−^ for 1 day and then in 10^−8^ M ClO_4_^−^ for another day”.

### 2.4. Sensors Calibration and ClO_4_^−^ Determination 

One-milliliter aliquots of 1.0 × 1^−1^–1.0 × 10^−8^ M ClO_4_^−^ solutions were transferred to 25 mL beakers containing 9.0 mL of 50 mM phosphate buffer solution of pH 5.5. The GC/ETH500/SWCNTs/ClO_4_^−^-ISEs is inserted into the solution in conjunction with a double junction Ag/AgCl reference electrode. The EMF readings were recorded and plotted as a function of log*a_ClO_*_4^−^_. The obtained calibration graph was used for all subsequent measurements of unknown ClO_4_^−^ concentrations.

For successful assessment of perchlorate using the presented method, GC/ETH500/SWCNTs/ClO_4_^−^-ISEs were applied for perchlorate assessment in commercial firework samples. Two firework shell samples were homogenized using an agate mortar and left to dry under vacuum for one hour at room temperature. An accurate amount of the powder (0.5–1.0 gm) was transferred to a 50 mL beaker and was dissolved in 50 mL of de-ionized bi-distilled water. The solution is then carefully heated at 60 °C on a water-bath for 5 min. After that, it was left to cool, filtered, and completed to 100 mL with de-ionized bi-distilled water. As mentioned above, the amount of perchlorate was potentiometrically measured. 

For comparison, determination of ClO_4_^−^ using ion chromatography (IC) was carried out. Typically 10 mL of the above final test solution was further diluted to 100 mL. Before the analysis, ~5 mL of the test solution was filtered and 100 μL aliquots were injected into the chromatographic column.

## 3. Results and Discussions

### 3.1. Performance Characteristics of All Solid-Contact Perchlorate ISEs

The electrochemical performance of the SC/ISEs was evaluated according to the IUPAC recommendations [58]. Validation of the presented assay method was also done. After a period of three months, the performance characteristics of the proposed SC/ISEs are given in Table 1. As shown in Figure 1, the GC/ETH500/SWCNTs/ClO_4_^−^-ISEs reveals excellent response performance over a linear range between 1.0 × 10^−2^ and 1.0 × 10^−6^ M with a Nernstian response of −56.0 ± 1.1 mV/decade (n = 6, *R^2^* = 0.9998) and detection limit of 1.8 × 10^−7^ M. 

The transduction mechanism of using SWCNTs is linked to the formation of an electrical double layer at the interface between the ISM and SWCNTs [59]. This interface acts as an asymmetric capacitor confirming that the adsorption of a lipophilic TDMA^+^ cation in ISM onto the SWCNTs can contribute to the electrical double layer formation [60]. The mechanism of ion-to-electron transduction is schematically shown in Figure 2. At the interface between ISM and SWCNTs solid-contact, the large surface area of the later can provide more sites for TDMA^+^ adsorption and then it can facilitate the conversion of the ionic signal to an electrical signal [60]. 

Indium (III)-porphyrin ionophore interacts with ClO_4_^−^ causing an increase of the coordination number of In^III^ central atom from 3 to 5 or 6. Binding of perchlorate and other anions with indium porphyrin are expected because the electron density on the central In^III^ atom varies by the extent of donation from the equatorial ligands. As reported before [61], In^III^-porphyrin can bind with perchlorate forming mono-and di-perchlorate anion at its axial position without further complexation with other anions. It appears that at the interface between the ISM and sample, ClO_4_^−^ ion binds selectively with central In^III^ in porphyrin ligand.

Using Equation (1), student’s (*t*) value was calculated from data obtained by repeated measurements (n = 6) of 5 μg/mL internal quality control (*IQC*) ClO_4_^−^ sample. The *t_exp_* was 0.912 at 95% confidence interval and compared with the theoretical value (*t* = 2.015). This indicates that the null hypothesis was held.*t_exp_* = [(*μ* − *x*)√*n*]/*σ_s_*(1)
where *μ* is the *IQC* sample concentration, *x* is the found experimental average concentration, *n* is the number of replicates (*n = 6*) and *σ_s_* is the standard deviation. All validation characteristics, such as accuracy, precision, within-day repeatability, between-days reproducibility and relative standard deviation were presented in Table 1. Precision (relative standard deviation (*RSD*) or the coefficient of variance (*CV*) of the method was checked by using six replicate measurements of 10 µg/mL of a quality control ClO_4_^−^ sample. The precision and accuracy of the used procedure were calculated using the following equations:
Accuracy, % = (*x*/*µ*) × 100
(2)

Precision (*RSD*), % = (*S*/*x*) × 100
(3)
where *x*, *µ*, and *S* are the average of the measured perchlorate concentration, the reference standard perchlorate concentration, and standard deviation, respectively. The relative standard deviation was calculated and found to be 1.6. The dynamic response time of the solid-contact electrode revealed a fast response time of <10 s. Elimination of the inner filling solution prefers the short time response of the solid-contact ISEs as previously reported [59,60].

Effect of pH on the potential response of GC/ETH500/SWCNTs/ClO_4_^−^-ISEs was tested. The potential-pH relations revealed no potential variation by more than that ± 1 mV within the pH range of 4.5–7.5. At pH < 3, hydronium ion (H_3_O^+^) along with the formation of H_2_ClO_4_^+^ ions were perhaps extracted in the membrane phase and then compete with perchlorate ion for the cationic site in the membrane. At pH > 8, severe interference from OH^−^ ions were probably compete with ClO_4_^−^ for In^III^-porphyrin chelate ion. This is in a good approval with that reported by other workers in which the potential response of some anion-ISEs based on metalloporphyrin is affected by the change of pH within the range of 3–8 [61,62]. From all of the above, 50 mM phosphate buffer background of pH 5.5 was chosen for all subsequent measurements.

### 3.2. Interfering Ions Effect

Selectivity of GC/ETH500/SWCNTs/ClO_4_^−^-ISEs over many common anions was potentiometrically evaluated by measuring the selectivity coefficients using the modified separate solutions method (MSSM) [55]. This method is used to remove the effect of the inseparable limit in sensitivity on the potential response of the ISE toward the distinguished ions. The recorded results are presented in Table 2. As can be seen from these results, the selectivity coefficient values of GC/ETH500/SWCNTs/ClO_4_^−^-ISEs are in a good agreement with those obtained by the liquid-contact ISE based on the same used ionophore [45]. With the exclusion of SCN^−^ ions, high concentration levels of other anions commonly present, have no effect on the potentiometric response of the sensors in presence of perchlorate ions. The order of selectivity was: ClO_4_^−^ > SCN^−^ > I^−^ > Cl^−^ > NO_2_^−^ > Br^−^ > NO_3_^−^ > CN^−^ > N_3_^−^ > S_2_O_3_^2−^ > CH_3_COO^−^ > S^2−^ > SO_4_^2−^ > PO_4_^3−^. Two possible mechanisms for the interaction of ClO_4_^−^ anion with In^III^-porphrin. According to neutral-carrier mechanism, ClO_4_^−^ is extracted from the aqueous medium into the membrane containing the neutral indium mono-perchlorate complex as a 6^th^ ligand for central In^III^ atom. This produces an octahedral negatively charged indium di-perchlorate complex. According to the mechanism of charged-carriers, interaction of perchlorate anion with indium-porphyrin charged molecule forms the neutral indium mono-perchlorate molecule and then the phase boundary potential is created. 

### 3.3. Short-Term Potential Stability 

Chronopotentiometry using current-reversed technique was used for short-term potential stability evaluation for the proposed sensors. As shown in Figure 3, the typical chronopotentiograms for the GC/ETH500/SWCNTs/ClO_4_^−^-ISEs and GC/ClO_4_^−^-ISEs were recorded in 1.0 × 10^−4^ M ClO_4_^−^ solution. According to the equation *ΔE*/*Δt* = *I/C* proposed by Bobacka [63], the potential drift (*ΔE*/*Δt*) is correlated with the implemented current (*I* = 10^−9^A) and the electrode low-frequency capacitance (*C*). Therefore, ISEs have a large capacitance (*C*) reveal low drift in the potential. The potential drift of the GC/ETH500/SWCNTs/ClO_4_^−^-ISEs was found to be 2.61 ± 0.7 µV/s, while GC/ClO_4_^−^-ISE revealed a potential drift 123 ± 2.4 µV/s. The evaluated low-frequency capacitances for the GC/ETH500/SWCNTs/ClO_4_^−^-ISEs and GC/ClO_4_^−^-ISE were found to be 383.2 ± 0.7 µF and 8.1 ± 0.3 µF, respectively. These results indicate that the introduction of ETH500/SWCNTs between the ISM and electronic conductor substrate can effectively enhance the potential stability of all-solid-state ClO_4_^−^-ISEs via increasing the low-frequency capacitance on the interface between the solid-contact material and ISM. 

### 3.4. Impedance Measurements

The impedance spectra of GC/ETH500/SWCNTs/ClO_4_^−^-ISEs and GC/ClO_4_^−^-ISEs were tested in 1.0 × 10^−4^ M ClO_4_^−^ solution to evaluate both high-frequency and charge-transfer resistances. In addition, double layer capacitances were also evaluated. As indicated in Figure 4, each ISE reveals a high-frequency semicircle, which represents the bulk resistance (*R_b_*) and geometric capacitance of the ISM. In the high-frequency pat, the resistance values for GC/ETH500/SWCNTs/ClO_4_^−^-ISEs and GC/ClO_4_^−^-ISEs were 0.34 ± 0.02 and 0.33 ± 0.04 MΩ, respectively. In addition, in the low-frequency part, the GC/ClO_4_^−^-ISEs reveals a larger semicircle than the one obtained in GC/ETH500/SWCNTs/ClO_4_^−^-ISEs. The low-frequency capacitance (*C_L_*) for GC/ETH500/SWCNTs/ClO_4_^−^-ISEs and GC/ClO_4_^−^-ISEs was *C_L_* = 27.6 ± 0.7 and 6.5 ± 1.2 µF, respectively. This indicates the existence of a high double layer capacitance (*C_L_*) and low charge transfer resistance at the interface between the sensing membrane and GC electrode.

### 3.5. Determination of ClO_4_^−^ in Commercial Fireworks Formulations

To test the validity of the proposed sensors, ClO_4_^−^ ions were determined in some commercial fireworks. About more than 50% of the constituents of these commercial fireworks are additives, so the response of GC/ETH500/SWCNTs/ClO_4_^−^-ISEs towards these additives was investigated. No noticeable interferences were found by the presence of 1000-fold excess of reducing agents such as sulfur and charcoal, binders such as dextrin and lactose, linseed oil as color brighten and aluminum flakes as regulators. As shown in Table 3, *F*-test showed no significant difference at 95% confidence level between means and variances of the proposed potentiometric technique and the standard ion chromatography for comparison. Determination of ClO_4_^−^ in some pure propellant powders of purity >99% was also carried out using the proposed perchlorate ISE. A shown in Table 4, the results obtained by the proposed sensor is in a close agreement and good reliability with this obtained by the ion chromatography method.

## 4. Conclusions

Simple and robust solid-contact ISE has been proposed for perchlorate determination. The fabrication of the sensor is based on the combination of using SWCNTs and the good adhesion ability revealed by ETH 500. As compared to GC/ClO_4_^−^-ISEs (CWEs), the proposed GC/ETH500/SWCNTs/ClO_4_^−^-ISEs revealed a significant enhancement in their potential stability. Moreover, the sensors introduced enhanced sensing characteristics including a broad linear range, fast response time, long-life span, and long-term stability. The sensors were used for the assessment of ClO_4_^−^ content in some fireworks and propellant powders. Validation of the method is carried out and the data obtained by the proposed method were compared with those obtained by the standard ion chromatographic method. The sensors revealed enhanced features over many of those previously reported in terms of robustness, ease of fabrication, selectivity, and accuracy. The sensors can be introduced in a flow system for continuous monitoring. Sample pretreatment is not required for perchlorate analysis using these proposed sensors.

## Figures and Tables

**Figure 1 sensors-19-02697-f001:**
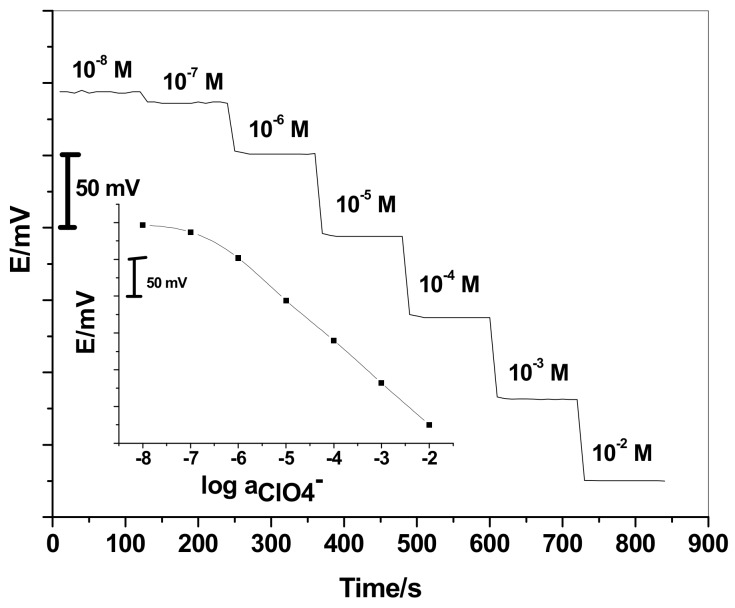
Potentiometric response of perchlorate based sensor (GC/ETH500/SWCNTs/ClO_4_^−^-ISEs)

**Figure 2 sensors-19-02697-f002:**
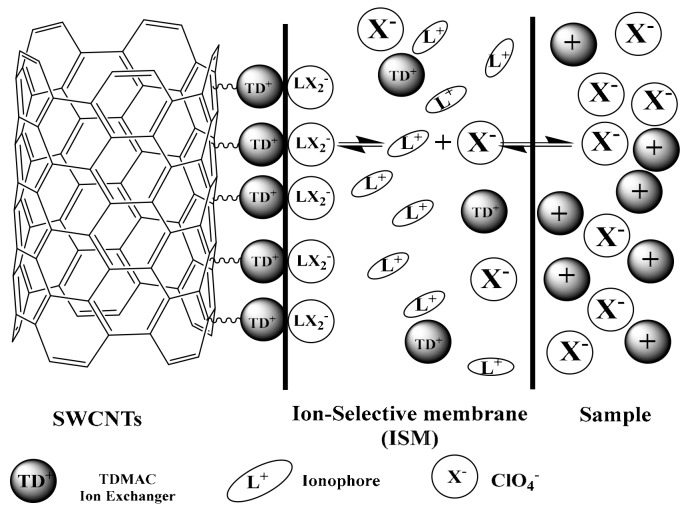
Schematic illustration of the transduction mechanism.

**Figure 3 sensors-19-02697-f003:**
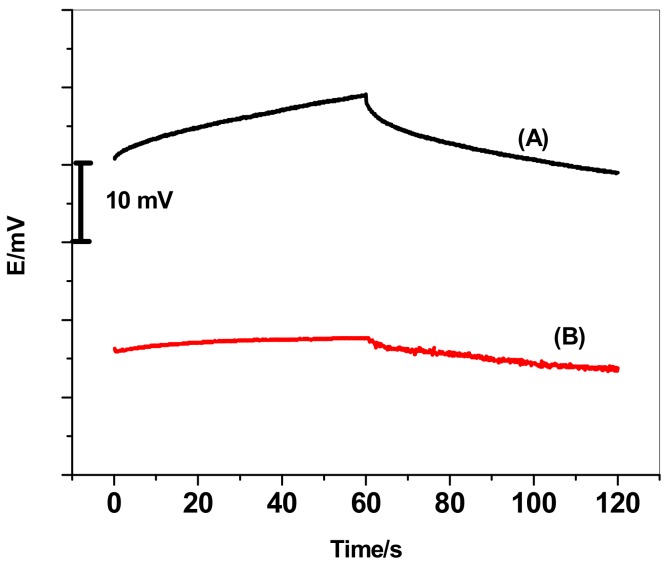
Chronopotentiograms (applied current: ± 1 nA for 60 s) for all-solid-state perchlorate ISE: (**A**) GC/ClO_4_^−^-ISE; (**B**) GC/ETH500/SWCNTs/ClO_4_^−^-ISEs.

**Figure 4 sensors-19-02697-f004:**
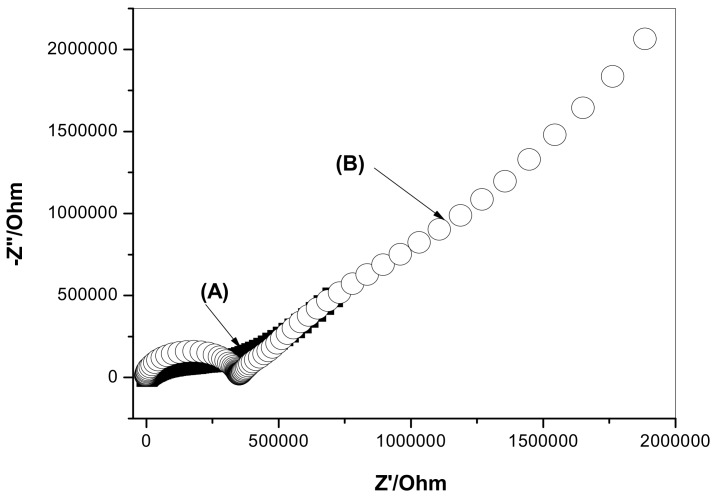
Impedance spectra for the proposed (**A**) GC/ETH500/SWCNTs/ClO_4_^−^-ISEs and (**B**) GC/ClO_4_^−^-ISEs.

**Table 1 sensors-19-02697-t001:** Performance characteristics of GC/ETH500/SWCNTs/ClO_4_^−^-ISE.

Parameter ^*^	GC/ETH500/SWCNTs/ClO_4_^−^-ISE
Slope, (mV/decade)	−56.0 ± 1.1
Correlation coefficient, (*r*^2^)	−0.9998
Lower detection limit, (M)	1.8 × 10^−7^
Linear range, (M)	1.07 × 10^−6^–1.0 × 10^−2^
Working acidity range, (pH)	4.5–7.5
Response time, (s)	<10
Life span, (week)	8
Precision, (%)	1.6
Accuracy, (%)	98.5
Standard deviation, (σ_mV_)	0.82

* Mean of six measurements.

**Table 2 sensors-19-02697-t002:** Selectivity values (log *K^pot^_ClO_4_^−^,j_*) for perchlorate solid-contact sensors.

Interfering Ion, *j*	GC/ETH500/SWCNTs/ClO_4_^−^-ISE *
SCN^−^	−0.9 ± 0.07
I^−^	−2.9 ± 0.5
Cl^−^	−3.3 ± 0.6
NO_2_^−^	−3.7 ± 0.7
Br^−^	−4.1 ± 0.4
NO_3_^−^	−4.2 ± 0.6
CN^−^	−4.5 ± 0.3
N_3_^−^	−4.6 ± 0.7
S_2_O_3_^2−^	−5.6 ± 0.4
CH_3_COO^−^	−6.1 ± 0.2
S^2−^	−6.5 ± 0.7
SO_4_^2−^	−7.2 ± 0.3
PO_4_^3−^	−7.8 ± 0.6

* Mean of three measurements.

**Table 3 sensors-19-02697-t003:** Potentiometric assessment of ClO_4_^−^ in some commercial firework samples.

Fireworks	[ClO_4_^−^] (%) ^a^				*F*-test ^b^
	Potentiometry	RSD, %	Ion Chromatography	RSD, %	
Sample 1	35.3 ± 1.2	3.4	31.2 ± 0.9	2.8	2.341
Sample 2	39.1 ± 1.7	4.3	35.7 ± 0.4	1.1	1.663
Sample 3	46.3 ± 2.2	4.7	42.1 ± 1.5	3.5	1.851

^a^ Average of 6 measurements. ^b^ Critical tabulated *F*-value (n = 6) = 5.05 at 95% confidence interval.

**Table 4 sensors-19-02697-t004:** Potentiometric assessment of perchlorate in some propellants.

Compound	[ClO_4_] (%) *		RSD, %
	Calculated	Found	
Urea perchlorate	62.0	61.3 ± 0.7	1.1
Hydrazine perchlorate	75.1	73.6 ± 1.5	2.1
Ethylenediamine perchlorate	62.0	60.4 ± 1.1	1.8
Ammonium perchlorate	84.7	81.2 ± 0.6	0.7

* Average of six measurements.

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
