# Peer review of "Single-Walled Carbon Nanotubes (SWCNTs) as Solid-Contact in All-Solid-State Perchlorate ISEs: Applications to Fireworks and Propellants Analysis"

_sensors, 2019, doi:10.3390/s19122697_

Round 1

Reviewer 1 Report

This work described by Saad S. M. Hassan and others is interesting for researchers working on potentiometric sensors. This article concerns discussion all-solid-state perchlorate ion-selective electrode and very interesting application  to fireworks and propellants analysis. The manuscript can be recommend for publication after completing the missing many information and improving several errors.

In detail:

1. Construction of the electrode: Do the authors use their own elaborated electrode constructions? The same system is used in another manuscript:

A simple approach for fabricating solid-contact ion-selective electrodes using nanomaterials as transducers Analytica Chimica Acta, 1 (2015), 291-296.

2. Validation of the electrode: (verse 276) . There is a lack of detailed description of electrode validation: how was accuracy, precision, repeatability, reproducibility and relative standard deviation designated and for how many sensors. Please specify in what solution , at which concentration , how long the measurements were carried out.

3. p. 3.2. Interfering ions effect (verse 208): Please explain,   why “…. the presence of ETH 500 can support the two mechanisms…..”. The Authors mention that the role of ETH 500 is better adhesion of the layer on glassy carbon.

In this method (MSSM) the sensors are conditioned in discriminated ions. But in p. 2.3. Authors write …”The ClO4‐‐ISEs were firstly conditioned in 10-3 M ClO4  for 1 day and then in 108 M ClO4 for another day. How were the electrodes prepared for measurement and how were the concentration of solutions ? How many repeating (e. g. for K= …±1.1) have been made ?

In Table 2,  instead of K there are values of log K. It should be (verse 209) “Selectivity values (log Kpot ClO4-,j) for perchlorate solidcontact sensors.”

4. Abbreviations used are inaccurate e.g. CWE (verse 115) the same as GC/ClO4- ISE (p. 3.3. and 3.4) the same as ClO4- ISE (p.3.5)  ?

5. Introduction (verse 48-52) : to cite  the reference of the electrodes in many analytical applications it should be given the latest review articles or book chapters too e.g.:

Eric de Souza Gil; Giselle Rodrigues de Melo, Electrochemical biosensors in pharmaceutical analysis Braz. J. Pharm. Sci. 46 (3) 2010

J. Lenik, Application of PVC in constructions of ion selective electrodes for pharmaceutical analysis, in: V. Kumar Thakur, M. Kumari Thakur (Eds), Handbook of Polymers for Pharmaceutical Technologies, Volume 2, Processing and Applications, Wiley Scrivener Publishing, 2015, pp.195-227.

J. Lenik, Cyclodextrins based electrochemical sensors for biomedical and pharmaceutical analysis -review , Current Medicinal Chemistry 24 (2017) 2359-2391

Sak-Bosnar, M., Madunić-Čačić, D., Grabarić, Z., Grabarić, B. Potentiometric Determination of Anionic and Nonionic Surfactants in Surface Waters and Wastewaters Handbook of Environmental Chemistry Vol 31, 2015, Pages 157-176

Yan, R., Qiu, S., Tong, L., Qian, Y.,  Review of progresses on clinical applications of ion selective electrodes for electrolytic ion tests: From conventional ISEs to graphene-based ISEs (Review) Chemical Speciation and Bioavailability  28,  1-4, ( 2016) 72-77

Dimeski, G., Badrick, T., John, A.S.  Ion Selective Electrodes (ISEs) and interferences-A review (Review), Clinica Chimica ActaVolume 411, Issue 5-6, 2 March 2010, Pages 309-317

Cuartero, M., Crespo, G.A. All-solid-state potentiometric sensors: A new wave for in situ aquatic research(Review) Current Opinion in Electrochemistry, 10,(2018),  98-106

The reference [25] not applicable control process , but quality control  criteria for ISE-s. please check it again.

Author Response

This work described by Saad S. M. Hassan and others is interesting for researchers working on potentiometric sensors. This article concerns discussion all-solid-state perchlorate ion-selective electrode and very interesting application to fireworks and propellants analysis. The manuscript can be recommend for publication after completing the missing many information and improving several errors.

In detail:

1. Construction of the electrode: Do the authors use their own elaborated electrode constructions? The same system is used in another manuscript: A simple approach for fabricating solid-contact ion-selective electrodes using nanomaterials as transducers Analytica Chimica Acta, 1 (2015), 291-296.

The reference is presented in line 106 as reference 57.

2. Validation of the electrode: (verse 276). There is a lack of detailed description of electrode validation: how was accuracy, precision, repeatability, reproducibility and relative standard deviation designated and for how many sensors. Please specify in what solution, at which concentration, how long the measurements were carried out.

All validation parameters of the constructed electrodes were clearly presented now in lines 170-186.

3. p. 3.2. Interfering ions effect (verse 208): Please explain,   why “…. the presence of ETH 500 can support the two mechanisms…..” The Authors mention that the role of ETH 500 is better adhesion of the layer on glassy carbon.

In fact, I did not find any previously reported work can support this statement. I found that ETH500 can affect the selectivity towards divalent anions for membranes based on anion-exchanger. So, I cannot prove that the presence of ETH 500 can support the two mechanisms and I removed this statement from the manuscript. But ETH500 has a great role for the good adhesion on GC substrate (Analytica Chimica Acta, 1 (2015), 291-296).

In this method (MSSM) the sensors are conditioned in discriminated ions. But in p. 2.3. Authors write …”The ClO4‐ISEs were firstly conditioned in 10-3 M ClO4 for 1 day and then in 10‐8 M ClO4 for another day. How the electrodes were prepared for measurement and how were the concentrations of solutions? How many repeating (e. g. for K= …±1.1) have been made?

As well known about MSSM, all sensors are conditioned in the less discriminated ions (i.e. 50 mM phosphate solution. For perchlorate determination, the sensors were firstly conditioned in 10-3 M ClO4 for 1 day and then in 10‐8 M ClO4 for another day. Log K values were of average of three measurements. This is indicated below table 2.

In Table 2, instead of K there are values of log K. It should be (verse 209) “Selectivity values (log Kpot ClO4-,j) for perchlorate solid‐contact sensors.”

It is corrected to Log K (i.e. line 218).

4. Abbreviations used are inaccurate e.g. CWE (verse 115) the same as GC/ClO4- ISE (p. 3.3. and 3.4) the same as ClO4- ISE (p.3.5)?

CWE abbreviation is corrected to GC/ClO4ISEs (line 116). GC/ClO4- ISE (p. 3.3. and 3.4) the same as ClO4- ISE (p.3.5) were corrected into GC/ETH500/SWCNTs/ClO4ISEs ( lines 122 and 256).

5. Introduction (verse 48-52): to cite the reference of the electrodes in many analytical applications it should be given the latest review articles or book chapters too e.g.:

All the mentioned references were added in the manuscript as:

Eric de Souza Gil; Giselle Rodrigues de Melo, Electrochemical biosensors in pharmaceutical analysis Braz. J. Pharm. Sci. 46 (3) 2010   ( ref. 31)

J. Lenik, Application of PVC in constructions of ion selective electrodes for pharmaceutical analysis, in: V. Kumar Thakur, M. Kumari Thakur (Eds), Handbook of Polymers for Pharmaceutical Technologies, Volume 2, Processing and Applications, Wiley Scrivener Publishing, 2015, pp.195-227. (ref. 30)

J. Lenik, Cyclodextrins based electrochemical sensors for biomedical and pharmaceutical analysis -review, Current Medicinal Chemistry 24 (2017) 2359-2391 (ref. 29).

Sak-Bosnar, M., Madunić-Čačić, D., Grabarić, Z., Grabarić, B. Potentiometric Determination of Anionic and Nonionic Surfactants in Surface Waters and Wastewaters Handbook of Environmental Chemistry Vol 31, 2015, Pages 157-176 (ref. 25)

Yan, R., Qiu, S., Tong, L., Qian, Y.,  Review of progresses on clinical applications of ion selective electrodes for electrolytic ion tests: From conventional ISEs to graphene-based ISEs (Review) Chemical Speciation and Bioavailability  28,  1-4, ( 2016) 72-77 (ref. 18).

Dimeski, G., Badrick, T., John, A.S.  Ion Selective Electrodes (ISEs) and interferences-A review (Review), Clinica Chimica ActaVolume 411, Issue 5-6, 2 March 2010, Pages 309-317 (ref. 19)

Cuartero, M., Crespo, G.A. All-solid-state potentiometric sensors: A new wave for in situ aquatic research(Review) Current Opinion in Electrochemistry, 10,(2018),  98-106 (ref. 24)

The reference [25] not applicable control process, but quality control criteria for ISE-s. please check it again

It is modified into quality control criteria (line 50).

Reviewer 2 Report

In the work a new sensor for perchlorate ion determination in fireworks and propellant was presented. The authors have a long experience on this topic, just read one of their works “A new liquid-membrane electrode for selective determination of perchlorate, Saad S.M. Hassan, M.M. Elsaied; Talanta Volume 33, Issue 8, August 1986, Pages 679-684” of the remote 1986.

The sensor shows better performance since the use of SWCNTs on an electrode has increased the specific exposed surface and therefore the reactivity to the ion with a reduction also in the response time.

The author characterized this new electrode clearly and absolutely exhaustive. Unfortunately the novelty can only be seen in the fabrication of the electrode.

According my opinion, in order to be published, authors should try to stress the points of novelty respect to their previous works, highlighting the progress made.

Author Response

In the work a new sensor for perchlorate ion determination in fireworks and propellant was presented. The authors have a long experience on this topic, just read one of their works “A new liquid-membrane electrode for selective determination of perchlorate, Saad S.M. Hassan, M.M. Elsaied; Talanta Volume 33, Issue 8, August 1986, Pages 679-684” of the remote 1986. The sensor shows better performance since the use of SWCNTs on an electrode has increased the specific exposed surface and therefore the reactivity to the ion with a reduction also in the response time. The author characterized this new electrode clearly and absolutely exhaustive. Unfortunately the novelty can only be seen in the fabrication of the electrode.

According my opinion, in order to be published, authors should try to stress the points of novelty respect to their previous works, highlighting the progress made.

I completely respect the reviewer’ s point of view, but as mentioned in the manuscript there are many points can clarify the novelty of this work  from that previously published by our co-workers such as:

1. The  fabrication of the electrode.

2. The  use of SWCNTS with InIII- porphyrine.

3. Better  detection limit and better selectivity.

4. Characterization of the proposed sensor using EIS and chronopotentiometric measurements.

5. Application  to fireworks and propellents.

6. Validation for the proposed method was verified.

7. Comparison of the proposed sensor with others previously reported was indicated in      lines 58-67.

Finally, I present my deep appreciation to reviewers for their valuable comments.

Round 2

Reviewer 1 Report

I accept the manuscript for publication.

Reviewer 2 Report

Status: Accepted

I appreciate the authors’ effort to justify what are the novelty points compared to their previous works. However, I still think that there is not a remarkable state of progress compared to what was published in 1986. Points 1 and 2 are practically the same, that is the electrode fabrication, the sensitivity curve has the same slope and also the final application to propellants is the same. So, in my opinion, the work lacks novelty.

The work is certainly well written, every aspect is addressed in depth, and if that's good enough for Editor, it can be published on Sensors.